# Rapid-Detection Sensor for Rice Grain Moisture Based on NIR Spectroscopy

**Lei Lin [1,2], Yong He [1,2], Zhitao Xiao [3], Ke Zhao [3], Tao Dong [1,2] and Pengcheng Nie [1,2,*]**

[1] College of Biosystems Engineering and Food Science, Zhejiang University, Hangzhou 310058, China; linlei2016@zju.edu.cn (L.L.); yhe@zju.edu.cn (Y.H.); dt2016@zju.edu.cn (T.D.)
[2] Key Laboratory of Spectroscopy Sensing, Ministry of Agriculture, Hangzhou 310058, China
[3] College of Information Engineering, Nanchang Hangkong University, Nanchang 330063, China; xzt541538701@163.com (Z.X.); zhaoke6805@126.com (K.Z.)
[*] Correspondence: npc2012@zju.edu.cn; Tel.: +86-0571-8898-2456

**Abstract:** Rice grain moisture has a great impact on th production and storage storage quality of rice. The main objective of this study was to design and develop a rapid-detection sensor for rice grain moisture based on the Near-infrared spectroscopy (NIR) characteristic band, aiming to realize its accurate and on-line measurement. In this paper, the NIR spectral information of grain samples with different moisture content was obtained using a portable NIR spectrometer. Then, the partial least squares (PLS) and competitive adaptive reweighted squares (CARS) were applied to model and analyze the spectral data to find the rice grain moisture NIR spectroscopy. As a result, the 1450 nm band was sensitive to the rice grain moisture and a rapid-detection sensor was developed with a 1450 nm light emitting diode (LED) light source, InGaAs photodiode, lens and filter, whose basic principle is to establish the relationship between the rice grain moisture and the measured voltage signal. To evaluate the sensor performance, rice grain samples with 13–30% moisture content were detected, the coefficient of determination $R^2$ was 0.936, and the sum of squares for error ($SSE$) was 23.44. It is concluded that this study provides a spectroscopic measuring method, as well as developing an effective and accurate sensor for the rapid determination of rice grain moisture, which is of great significance for monitoring the quality of rice grain during its production, transportation and storage process.

**Keywords:** NIR spectroscopy; rice grain moisture; 1450 nm; rapid-detection sensor

## 1. Introduction

Rice is a staple cereal crop in many countries, and quality assessment and process control is very important for rice grain due to the production seasonality and the huge demand [1,2]. The variation of moisture level not only affects the content of starch, fat and protein, but also affects the quality of rice grain to a certain extent [3–5]. Therefore, it is an indispensable prerequisite for safe crop storage to determine the rice grain moisture accurately and quickly. Generally, the method for determining moisture content can be divided into direct methods and indirect methods. The precision of direct methods is extremely high, but the complicated and time-consuming operation cannot meet the demands of effective and non-destructive determination. Thus, direct methods, represented by the oven-drying method, are often used as the standard to evaluate the precision of indirect methods.

In recent years, researchers have put forward several indirect methods for rapid and non-destructive measurement of rice grain moisture or other cereal crops [6–9]. Commonly used methods include the resistance method, capacitance method, spectroscopic method and dielectric loss angle method. Kok et al. [10] put forward a moisture detection method for single rice grains using a

slim open-ended coaxial probe, which was suitable for the nondestructive measurement of moisture values in the rice grains ranging from 9.5% to 26%. Li et al. [11] proposed a method for measuring grain moisture content (13–26%) based on dielectric loss tangent and the maximum error was less than 0.5%. Wirot et al. [12] designed a paddy rice moisture content meter based on the principle of electrical capacitance properties and the detection accuracy reached 97.83%.

Compared with the detection methods mentioned above, NIR spectroscopy has emerged as a more viable means, incorporating the advantages of real time, high precision and non-destructive detection [13]. The information of NIR spectra is mainly derived from O-H, C-H, N-H and other hydrogen-containing groups' internal vibration and absorption in overtone and combination bands. Furthermore, the O-H bond, the only bond in water, has several characteristic absorption peaks in the NIR region [14,15].

Hence, the moisture content of some seeds and grains can be detected effectively and non-destructively using NIR spectroscopy. Natsuga et al. [16] determined multiple physicochemical properties of rice grain using a visible and NIR reflectance spectrometer. The results of correlation coefficient of the validation set using partial least squares (PLS) with full cross-validation was higher than 0.8. Torbjörn et al. [17] measured the moisture content (3–34%) of single seed samples and bulk seed samples of Scots pine using NIR spectroscopy. PLS and ordinary least squares (OLS) regression were applied to deal with the spectral data and the prediction accuracy in different spectral range was evaluated. The results suggested that the lowest prediction error was 0.8% for bulk seed samples in the 850–1048 nm spectral range modeled by PLS. András et al. [18] monitored the maturation process of wheat seed by NIR spectroscopy. The results pointed out several characteristic bands of water, carbohydrates and proteins, where Water I (1890–1920 nm) and Water III (1150–1165 nm) bands were more sensitive to the variation of water in the maturing wheat seed than the Water II (1400–1420 nm) band, while all three water absorption bands could be used to monitor the changes of water and define the development stage of the wheat seed. Cem et al. [19] applied NIR spectroscopy to analyze the protein, oil, carbohydrate and ash content in maize grain. The results suggested that protein content could be successfully estimated ($R = 0.990$ for multiple linear regression (MLR), $R = 0.987$ for PLS, while carbohydrate ($R = 0.801$ for MLR, $R = 0.755$ for PLS), oil ($R = 0.823$ for MLR, $R = 0.723$ for PLS) and ash ($R = 0.926$ for MLR, $R = 0.810$ for PLS) required more studies to obtain better prediction results. Similarly, Fassio et al. [20] evaluated the potential of NIR spectroscopy for predicting the nutritive value of grain corn with high moisture content. The coefficients of determination in the calibration of dry matter, acid detergent fiber, in vitro organic matter digestibility, pH and ammonia nitrogen were all higher than 0.9. Jasper et al. [21] developed a single-kernel NIR instrument for predicting maize grain attributes, where crude protein content and kernel mass were more predictable than tryptophan, lysine and oil. Although many scholars have applied NIR spectroscopy for seed and grain moisture detection, the studies have mainly been based on the NIR spectrometers for laboratory use, consisting of complex optical path structure and not suitable for application in a farmland environment.

Therefore, late long-grain nonglutinous rice with a wide planting area was selected as the research object. The main objective of this paper is to present a moisture detection method for rice grain based on NIR spectroscopy characteristic band, as well as to design and develop a rapid-detection sensor based on a single-band 1450 nm. As a result, a portable, low-cost and high-precision sensor for measuring rice grain moisture was developed, which is of great significance for enhancing detection efficiency and monitoring of rice grain quality in real time.

## 2. Materials and Methods

### 2.1. Sample Preparation

For this study, late long-grain nonglutinous rice grown in Zunyi city (Guizhou province, China) was chosen as the experimental sample. The original rice grain moisture was relatively low, since the samples had been sun-dried before experiment. The sample preparation process was as follows. First,

the rice grain samples were soaked in water at 40–50 °C for 10 hours. Second, the samples were placed into a 25 °C thermostat for 5 hours; thus, the samples were able to absorb the moisture adequately, and the moisture content of samples was the highest at this point. Third, the samples were dried in an oven of 40 °C. To obtain rice grain of a gradient of moisture level, the drying time was set for 9 gradients, including 0 min, 5 min, 15 min, 30 min, 50 min, 75 min, 105 min, 145 min, and 195 min. For each drying time, ten samples (50 g each) were randomly selected and divided into duplicates (half for true moisture determination and half for NIR spectra acquisition). A total of 90 samples were prepared. Additionally, the true rice grain moisture was determined by the 105 °C constant weight method from the national standard GB5497-1985 (Determination of moisture content of grain and oil). Rice grain moisture is this paper is expressed as % of dry weight.

## 2.2. NIR Instrument and Spectra Acquisition

The spectral information of rice grain samples was collected by a NIR optical spectrum instrument (NIRez, Isuzu Optics Corp, Shanghai, China). This instrument is reflective, with two integrated tungsten halogen lamps, whose collection range is 950–1650 nm; optical resolution is 10 nm and the signal-noise ratio is 5000:1 in a one-second scan. The spectrometer can measure the penetration and reflectivity of the measured object. The spectrometer itself carries a light source that can be placed in the photosensitive window to directly obtain the spectrum. The reflectance spectrum is directed through the collimator to the sensor. There is only one sensor pixel and when the scanning test works, it can acquire the spectrum at different positions and at different times. When the spectrometer works in reflective mode, the light is acquired through a lens coupled to the spectrometer. Prior to spectral determination, the instrument was turned on and preheated for 15 min to its operating conditions. Then the blackboard and whiteboard correction was conducted. During the spectral acquisition process, the 90 rice grain samples prepared as described in Section 2.1 were loaded into a hollow cylinder with a diameter of 30 mm in sequence, and the corresponding full NIR spectral information was collected using this NIR optical spectrum instrument. The scan range was 950–1650 nm and the obtained spectrum for each sample was the average of three scans.

## 2.3. Chemometric Analysis

In this paper, partial least squares (PLS) and competitive adaptive reweighted squares (CARS) were used to model and analyze the NIR spectral data, as well as to determine the characteristic waveband sensitive to rice grain moisture. The Polynomial fitting Model was applied to fit the mathematical relationship between the reflected voltage value and the true moisture content of rice grain samples. In this study, data analysis was based on MATLAB R2014a (The MathWorks, Inc., Natick, MA, USA, 2014).

### 2.3.1. Partial Least Squares

Partial least squares (PLS) is the most widely used regression modeling method in spectral data analysis due to its flexibility and reliability in dealing with redundant spectral data [22]. It takes advantage of the variables containing the most comprehensive information, as well as identifying noise by decomposing and filtering the spectral data, and is then able to explore the linear combination of spectral data and the chemical compositions [23]. When PLS is applied to dealing with the spectral data, the spectral matrix is first decomposed, and the main principal component variables are obtained; then, the contribution rate of each principal component is calculated. The flexibility of PLS makes it possible to establish a regression model in cases where the number of samples is lower than the number of variables [24]. In this study, the PLS model was established with the spectral data as *X* (the number of the spectra original variables was 350) and the measured moisture content of rice grain as *Y*, whose best principal factor was determined by root mean square error of cross validation (RMSECV).

### 2.3.2. Competitive Adaptive Reweighted Squares

Competitive adaptive reweighted sampling (CARS) is a feature wavelength selection algorithm which imitates the principle of "Survival of the fittest" in Darwin's evolution theory. This algorithm adopts the idea of Monte Carlo sampling or the random sampling method, which selects part of the samples to conduct PLS modeling and repeatedly performs hundreds of modeling iterations [25]. In the process of selecting wavelength variables, the same variable set was repeatedly tested, and CARS remains the wavelength variable with the large absolute value of PLS regression coefficient and removes the wavelength variable with the lowest absolute value of PLS regression coefficient; thus, a series of subsets of wavelength variables is obtained [26]. Additionally, CARS calculates the RMSECV value of the model and selects the minimum value corresponding to the subset of variables for the optimal subset of variables.

### 2.3.3. Single Linear Regression

Single linear regression (SLR) is a statistical analysis method which uses regression analysis in mathematical statistics to determine the quantitative relationship between the two variables. In SLR, data are modeled by linear prediction function, and unknown model parameters are estimated by data. The most commonly used linear regression model is that the conditional mean of $Y$ with given $X$ value is an affine function of $X$ [27].

### 2.3.4. Polynomial Fitting Model

Polynomial fitting is a kind of function approximation method, which fits out the relationship between the independent variable $x$ and the dependent variable $y$ through mathematical calculation principles [28]. This data analysis method does not require the approximate function to be the same as the true value at each node; it only requires that the approximate function can reflect the basic trend of the given data point and a sense of "infinite approximation" as much as possible. The sum of squares for error (*SSE*) was used to indicate the stability of the model as follows.

$$SSE = \sum_{i=0}^{m} [p(x_i) - y_i]^2 \tag{1}$$

where $p(x_i)$ was the predicted moisture content, and $y_i$ was the true moisture content. $I = 1, 2 \ldots m$.

### *2.4. Variable Partition Method*

SPXY, the method of choosing the calibration sample, was put forward on the basis of KS methods by Galvao et al. [29]. The basic principle is that spectrum and concentration variables are considered at the same time to calculate the distance of the samples, the distance formula is as follows:

$$d_{xy} = \frac{d_x(i, j)}{max_{i,j \in (1,z)}[d_x(i, j)]} + \frac{d_y(i, j)}{max_{i,j \in (1,z)}[d_y(i, j)]}, i, j \in [1, z] \tag{2}$$

In the formula, $d_x(i, j)$ is based on spectral characteristic parameters for the calculation of the distance between the samples, while $d_y(i, j)$ is based on concentration characteristic parameters for the calculation of the distance between the samples—which causes the sample to have the same weightiness in both spectrum space and concentration space—divided by their corresponding maximum standardization, respectively. $z$ is spectral space.

### *2.5. Model Evaluation*

In this paper, the model performance was evaluated by the coefficient of determination ($R^2$) and the root mean square error (*RMSE*). The highest $R^2$ and the lowest *RMSE* were considered to

indicate the best performance. In this paper, $R_c^2$ and $R_p^2$ represent the coefficient of determination of the calibration set and the prediction set, respectively. In addition, *RMSEC* and *RMSEP* represent the root mean square error of the calibration set and the prediction set, respectively [30].

## 3. Results and Discussion

### 3.1. Characteristic Band Selection and Sensor Design

3.1.1. NIR Spectra Analysis and Characteristic Band Determination

For the effective determination of moisture content in rice grain samples, it might be necessary to reduce the full spectra from 950–1650 nm to a few characteristic bands according to the curvilinear trend and water absorption peaks. The raw NIR spectra of the 90 rice grain samples with different moisture content obtained in Section 2.2 are shown in Figure 1.

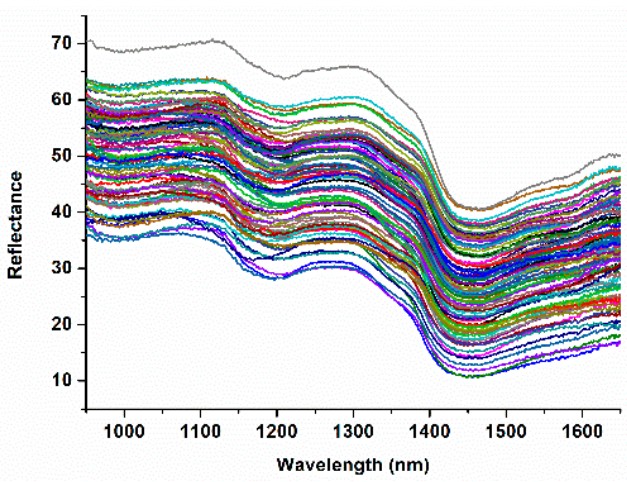

**Figure 1.** The NIR spectra of rice grain with different moisture content.

According to Figure 1, both ends of the spectra showed obvious noise caused by the decreasing efficiency of the diffraction grating towards the edges. Beyond that, there are two distinct absorption peaks in the NIR spectra of rice grain ranging from 1100–1250 nm and 1350–1550 nm. More specifically, the spectral valley appeared around 1450 nm (first overtone of O-H stretching) [20,31].

To explore the model performance of 950–1650 nm, 1100–1250 nm, 1350–1550 nm and 1450 nm, respectively, chemometric analysis (PLS, CARS, SLR) was applied to model and analyze the spectral data. In addition, the SPXY (sample set partitioning based on joint x-y distance) method was used to divide the rice grain samples into two parts, including 60 samples in calibration set and 30 samples in the prediction set. Also, the leave-one-out method was used for cross validation. Table 1 summarizes the statistics description of moisture content in 90 rice grain samples and the model performance of those three bands is shown in Table 2.

**Table 1.** Statistical description of the moisture content in rice grain samples.

| Set | Number | Min (%) | Max (%) | Mean (%) | SD [a] (%) |
|---|---|---|---|---|---|
| **Calibration set** | 60 | 10.46% | 27.88% | 20.26% | 5.81% |
| **Prediction set** | 30 | 10.27% | 25.98% | 19.18% | 5.34% |
| **Total** | 90 | 10.27% | 27.88% | 19.9% | 5.62% |

[a]: SD (standard deviation).

**Table 2.** The model performance of three bands by PLS and CARS.

| Wavebands | The Number of Variables | Algorithm | Calibration Set | | Prediction Set | |
|---|---|---|---|---|---|---|
| | | | $R_c^2$ | [b] RMSEC (%) | $R_p^2$ | RMSEP (%) |
| 950–1650 nm | 350 | PLS | 0.989 | 0.57 | 0.970 | 0.8877 |
| | | CARS | 0.973 | 0.83 | **0.977** | 0.93 |
| 1100–1250 nm | 72 | PLS | 0.938 | 1.32 | 0.844 | 2.39 |
| | | CARS | 0.865 | 1.94 | 0.883 | 2.12 |
| 1350–1550 nm | 105 | PLS | 0.930 | 1.53 | 0.923 | 1.46 |
| | | CARS | 0.943 | 1.52 | 0.917 | 1.52 |
| 1450 nm | 1 | SLR | 0.845 | 1.51 | 0.832 | 1.68 |

[b]: PLS (Partial Least Squares); CARS (Competitive Adaptive Reweighted Squares); RMSEC (root mean square error of the calibration); RMSEP (root mean square error of the prediction set).

Figure 2 and Table 2 show the model results of 950–1650 nm, 1100–1250 nm,1350–1550 nm and 1450 nm of rice grain modeled by PLS, CARS and SLR. As a result, the 950–1650 nm, containing the most comprehensive information, achieved the best prediction effects ($R_p^2$ = 0.970 for PLS and $R_p^2$ = 0.977 for CARS), while the prediction accuracy for 1100–1250 nm was the worst ($R_p^2$ = 0.844 for PLS and $R_p^2$ = 0.883 for CARS). For 1350–1550 nm, better prediction accuracy ($R_p^2$ = 0.923 for PLS and $R_p^2$ = 0.917 for CARS) was obtained than for 1100–1250 nm, indicating that 1350–1550 nm was more sensitive to rice grain moisture in the NIR region. Based on Figure 1, 1450 nm was determined as the optimal wavelength. Therefore, we established a regression model based on the near-infrared spectral absorption intensity of the 1450 nm band and rice grain moisture shown in Figure 3. Compared with the results by Cem et al. [19] and Fassio et al. [20], the prediction effects were greatly increased in our research. The results showed that there was a good regression between the near-infrared spectral absorption intensity of the 1450 nm band and the accuracy of rice grain moisture ($R_p^2$ = 0.832), indicating that the strong compression of *X* to one dimension in 1450 nm did not noticeably influence the prediction accuracy. Therefore, 1450 nm could be determined as the characteristic waveband for developing the rapid-detection sensor, with similar results being reported by other authors [21,32]. In summary, the LED light-emitting diode with 1450 nm wavelength was chosen as the light source for the detection sensor in this paper, the emission wavelength of which was also in the range of 1350–1550 nm.

### 3.1.2. The Principle of NIR Optical Detection

The detection principles of conventional NIR detection instruments typically include the transmission method and the reflection method [33]. The transmission method is generally used to detect transparent liquid, but the rice grain samples are irregular and thick in shape. Hence, the transmission method was not suitable in this case, and the reflection method was adopted. In this paper, the optical detection principle of the sensor probe was mainly based on the following. When the light source irradiates the rice grain samples, the intensity of the reflected light signal that is collected by photosensitive components directly reflects the moisture content in the rice grain samples [34]. In addition, the application of a high-pass filter can reduce the noise signal interference and the focusing lens can enhance the reflection signal strength, as well as improve the instrument detection performance.

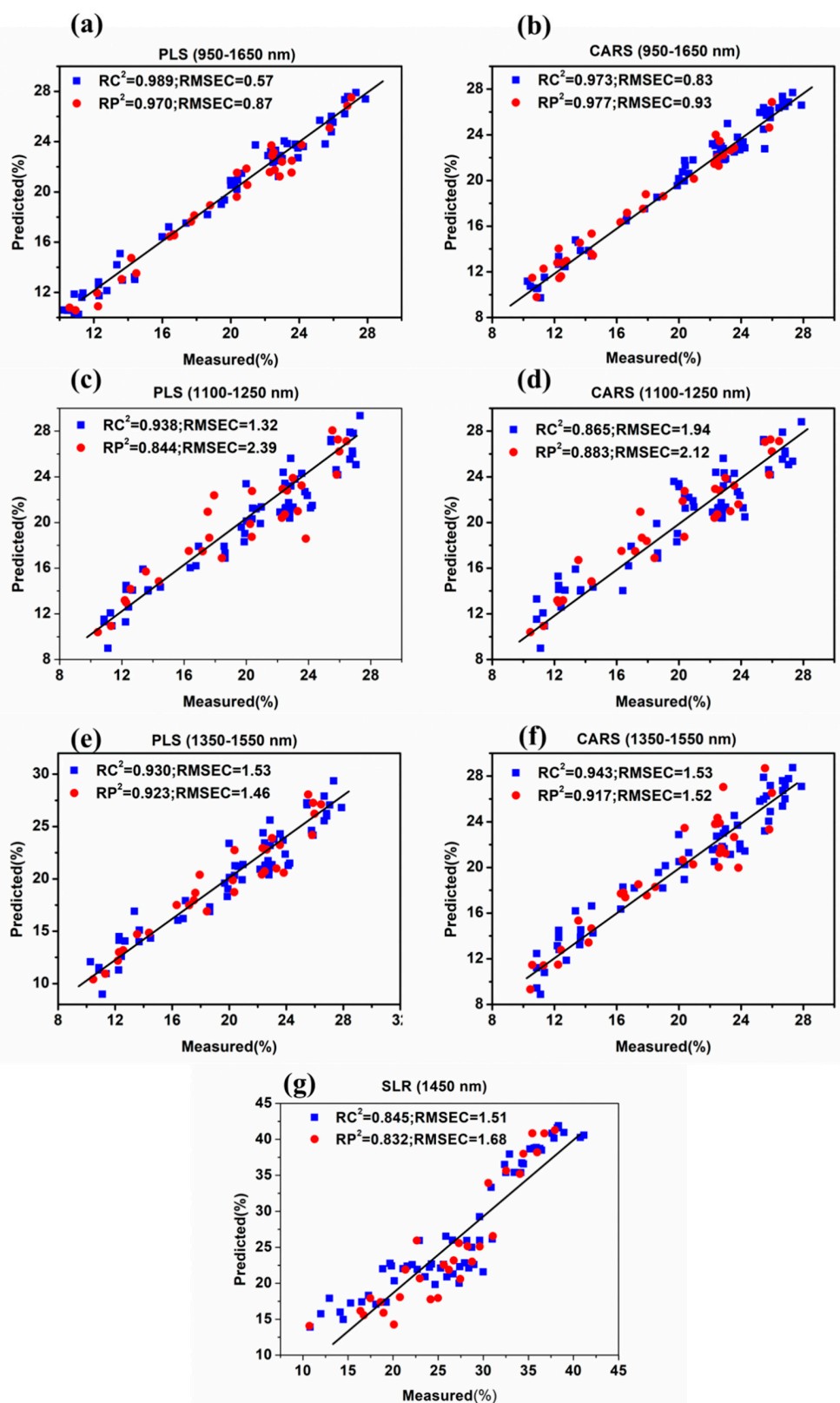

**Figure 2.** The model performance of 950–1650 nm modeled by Partial Least Squares (PLS) (**a**) and Competitive Adaptive Reweighted Squares (CARS) (**b**); The model performance of 1100–1250 nm Near-infrared spectroscopy (NIR) modeled by PLS (**c**) and CARS (**d**); The model performance of 1350-1550 nm NIR spectra modeled by PLS (**e**) and CARS (**f**). The model performance of 1450 nm NIR spectra modeled by SLR (**g**).

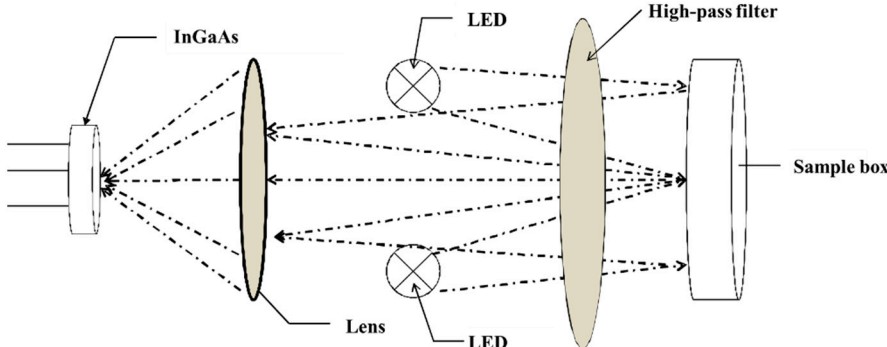

**Figure 3.** The schematic diagram and components of the sensor probe.

*3.2. Sensor Structure Design*

### 3.2.1. The Optical Design of the Sensor Probe

Figure 3 displays the structure view of the sensor probe in this moisture detection sensor. The entire sensor probe mainly consists of a fully symmetrical case, two light sources, a high-pass filter, a lens and a photodiode. The upper and lower shells of the sensor probe were printed by 3D printing technology, and an InGaAs photodiode with the dark current property was adopted, whose response range was from 800 nm to 1700 nm. Given that the maximum saturation current generated by photodiode sensitivity was only a few milliamperes (AD620), it was necessary to amplify the weak signal. Therefore, the sensor probe also integrated an I-V conversion amplifier which could convert a weak current signal to a larger voltage signal.

### 3.2.2. The Hardware Design

The hardware module of this rapid-detection sensor can be divided into three main parts: system control, signal acquisition and functional circuits. More specifically, the system control part includes the central processing unit (CPU) main control circuit, Ethernet circuit, time-sharing signal acquisition control circuit, human-machine interface circuit and LED constant current source drive circuit. The signal acquisition part includes temperature and humidity digital acquisition circuits, a photoelectric signal acquisition circuit, and an Analog Signal to Digital Signal (AD) signal acquisition circuit. In addition, the functional circuits include a power circuit and a flash data storage circuit.

Consistent with the principle of moisture content determination using NIR spectroscopy, the core portion of hardware was designed as follows. The linear predictive coding (LPC) 1768 chip equipped with the Advanced RISC Machines (ARM) Cortex M3 (NXP Semiconductors, Eindhoven, Netherlands) was selected as the core CPU, which contains rich on-chip peripherals and up to 512 KB Read Only Memory (ROM) and 64 k Synchronous Dynamic Random Access Memory (SDRAM). On this basis, it can store the detected data with flash memory chip or external expansion flash storage chip. Moreover, the voltage signal, collected by sensor prove and dealt with signal conditioning, was conducted with AD conversion by the 16-bit AD conversion chip AD7705. The converted data is related to the corresponding moisture content of rice grain samples through the calibration model and displayed through the DOG12864 LCD. In addition to displaying the moisture content of rice grain, this sensor is capable of model selection, data viewing, data transmission and other functions. Furthermore, the hardware platform also includes a signal conditioning circuit and an LED loaded constant current source driver circuit. The specific hardware block diagram is shown in Figure 4, while the sensor probe and internal hardware of this sensor is shown in Figure 5.

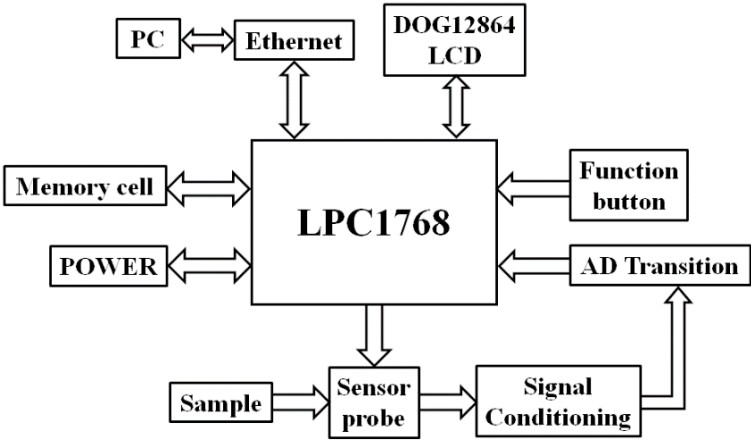

**Figure 4.** The block diagram of hardware design.

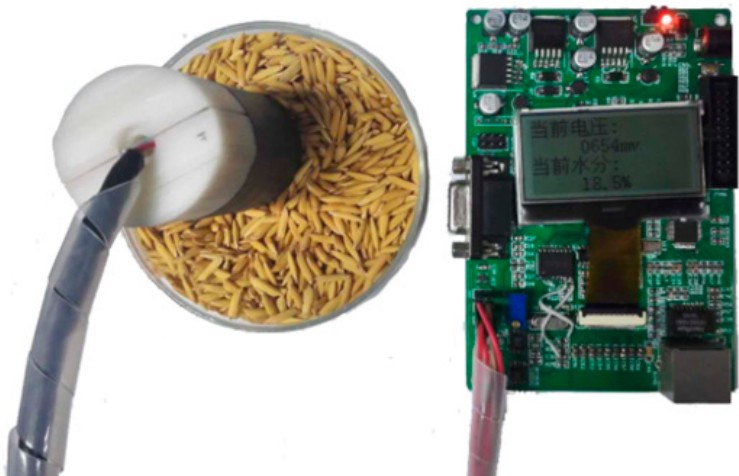

**Figure 5.** The sensor probe and printed circuit board.

### 3.2.3. The Software Design

The software system of moisture content determination of rice grain was programmed with a modular programming concept, mainly comprising a sensor probe control program, a rice grain moisture calculation program, an Ethernet control program, and a personal computer (PC) remote control program. More specifically, the UCOS (Micro-Controller Operating System) operating system was transplanted in this program, where the task functions of each function module were created through the main program of the system to realize functions such as sample measurement and data processing. Additionally, the task function module was divided into the following main sections: AD data acquisition module; Ethernet Transmission Control Protocol (TCP)/Internet Protocol (IP) data transmission module; model calculation module; Liquid Crystal Display (LCD) display module; function key selection module and FLASH memory data read-write module. Figure 6 presents the flowchart of this program.

Based on Figure 6, the program initializes each function module first and creates the function module task through the operating system when powered on. Once the scan button is pressed, the program will jump to the corresponding task according to the specific key instruction. When the assigned task is completed, all the tasks except the key scan task are suspended, and the above steps will be performed again until the scan button is pressed again. Furthermore, in addition to remote data reading, the instrument supports real-time remote/local PC control functions, which are realized on the basis of communication protocols established by the PC host computer. In general, this program supports the main functions of data processing (data acquisition, AD sampling, model establishment,

data storage and display), data view and data transmission, which is in accordance with the hardware design.

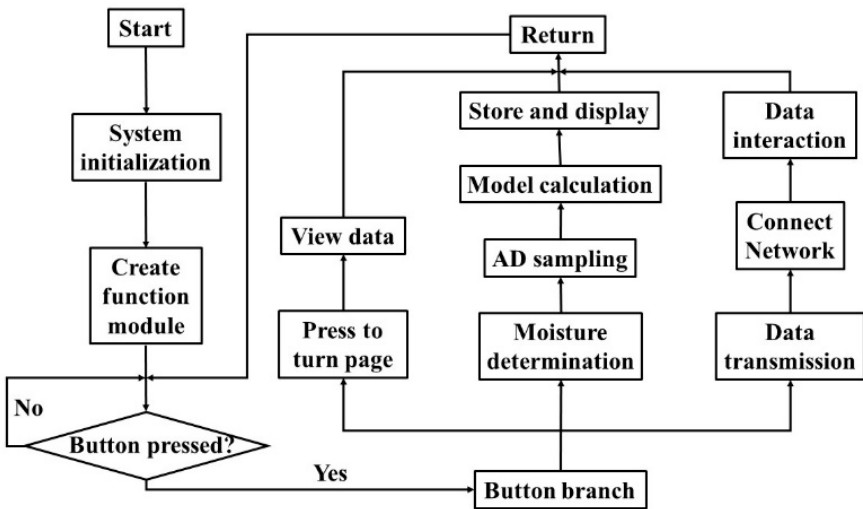

**Figure 6.** The flowchart of the software program.

### 3.3. Prediction Model Establishment

Test Preparation

To evaluate the performance of the designed rapid-detection sensor, 100 rice grain samples with different moisture content were prepared according to the sample preparation method described in Section 2.1. Among them, the calibration set contained 70 samples, while the prediction set contained 30 samples. Figure 7 displays the detection process of rice grain moisture using the designed rapid-detection sensor. In this paper, polynomial fitting was applied to fit the mathematical relationship between the reflected voltage value and the true moisture content of rice grain samples. The relationship between sensor voltage and obtained rice grain moisture was established as shown in Figure 8, and the formula was as shown below.

$$p(x) = -0.00005x^2 + 0.018x + 26.689 \tag{3}$$

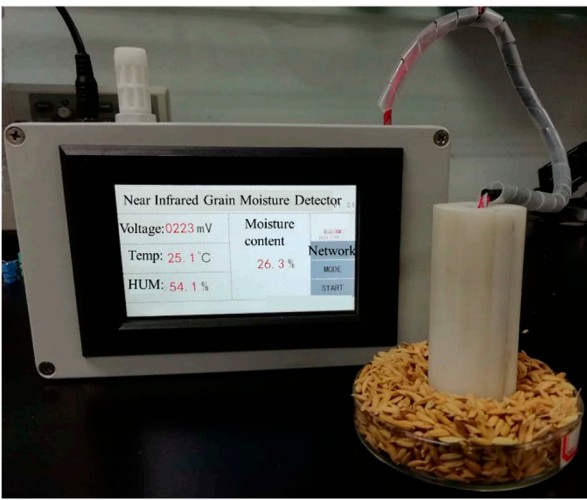

**Figure 7.** The detection process for test the sensor performance.

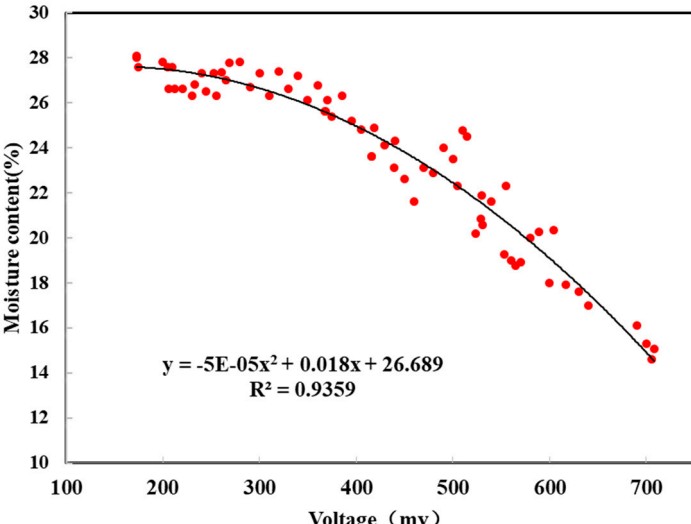

**Figure 8.** The relationship between sensor voltage and obtained rice moisture.

In this formula, $x$ is the voltage in mv, y is the moisture content in %. It can be seen from the figure that the sensor voltage has a good correlation with rice moisture. To verify the stability and reliability of the model, the moisture prediction model was transplanted into the software system of detection sensor. 30 rice grain samples (moisture content: 13–30%) were used to test the predictive ability of the model. The rice grain samples were detected by the designed detection sensor and compared with the true moisture content. In addition, the linear fitting relationship between the predicted moisture content and the true moisture content was obtained and shown in Figure 9.

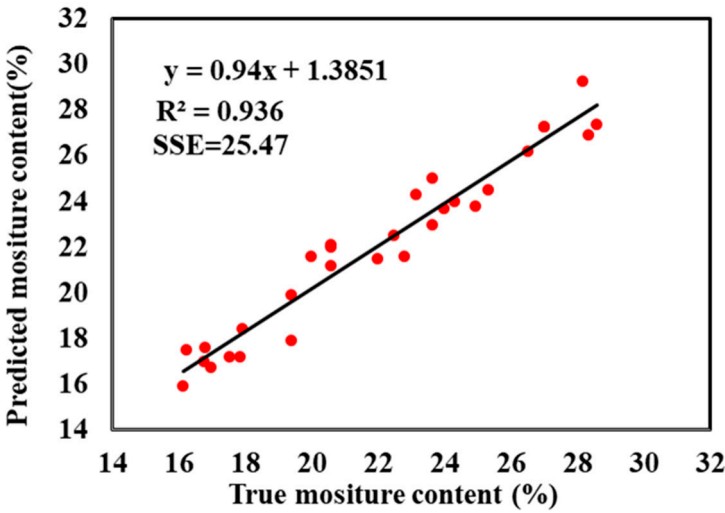

**Figure 9.** The linear fitting result between the detected moisture content and the true content.

As a result, by transplanting the prediction model of rice grain moisture and NIR reflection electrical signals into the system software program of the detection sensor, the designed sensor achieved good detection accuracy ($R^2 = 0.936$, SSE = 23.44) with respect to rice grain samples with 13–30% moisture content. Furthermore, no pretreatment method was applied in this experiment, which thus meets the requirements of rapid and accurate detection. Thus, in order to expand the moisture detection gradient of rice grain and enhance the detection accuracy in further studies, the number of rice grain samples can be increased to obtain more spectral information and optimize the moisture prediction model.

## 4. Conclusions

In this paper, a moisture prediction model of rice grains was established and a rapid-detection sensor of the moisture content in rice grain was designed and developed. An LED with 1450 nm was determined as the light source and the corresponding hardware circuit and software system was designed. The coefficient of determination $R^2$ was 0.936 and the sum of squares for error (*SSE*) was 25.47 when detecting the rice grain with 13–30% moisture content. Therefore, the proposed method based on NIR single-band measurement of the moisture content in rice grain achieved good results, which satisfied the practical production demands as well.

However, there are still several shortcomings of this sensor that need further research and improvement. First, we only studied a grain moisture detection model. As for the stability and transitivity of the model, further experimental exploration and verification are needed. Second, this sensor used a photodiode for optical-electrical signal conversion, while the output signal of photodiode was influenced by the measurement environment temperature and humidity. Thus, the impact of measuring temperature and humidity on the measurement results could be considered in further study, which might improve the detection accuracy to a certain extent.

**Author Contributions:** Conceptualization, L.L., Y.H., Z.X. and P.N.; Experiment, L.L., Z.X. and T.D.; Methodology, L.L.; Formal Analysis, Y.H., Z.X., K.Z. and P.N.; Data Curation, L.L., Y.H., and S.L.; Writing—Original Draft Preparation, L.L.; Writing—Review & Editing, L.L., Y.H., Z.X., K.Z., T.D. and P.N.

**Funding:** This research was supported in part by national key research and development plan (2018YFD0700704).

**Conflicts of Interest:** The authors declare no conflict of interest.

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
