# Peer review of "Rapid-Detection Sensor for Rice Grain Moisture Based on NIR Spectroscopy"

_applsci, doi:10.3390/app9081654_

Round 1
Reviewer 1 Report
This research manuscript described a homemade NIR instrument with potential suitability for in field analysis of rice moisture content, which is of great significance to help protect the safety and quality of rice grains in the farm. Although the experiments are well-designed and the results were discussed properly, there are several things the authors need to fix and/or address before it can be published.
1. Line 94, change the “different gradients of rice grain moisture” to “rice grain of a gradient of moisture level”.
2. Line 97, change “divided in duplicate” to “divided into duplicate”
3. Line 98, change “there were 90 samples in total” to “A total of 90 samples were prepared”. This should be a separate sentence.
4. Line 122, remove “In this study”
5. Line 131, change “then” to “and then”
6. Section 2.3, the author should indicate the specific method used for cross-validation and selection of samples for prediction in the Materials and Methods section.
7. Line 187-188, table 2 and figure 2. In the text, the author mentioned the model performance of the three bands, but the figure 2 and table 2 only showed the results of the chemometric models for the two bands ranging 1100-1250 nm and 1350-1550 nm. Please correct this.
8. Lin 194-203, how is your results compare with available publications? Please include some discussion into this paragraph.
9. Figure 3 and line 206-213, could you please explain why a quadratic regression is fitted instead of a linear regression? Could this result in overfitting of the model? Also, as indicated, the PLSR model using the full spectrum provided the best quantification ability, why not use the model of the full spectra for the further application?
10. Figure 4, please add the path of the laser beam and the reflected beam in the figure.
11. Figure 8, please translate the Chinese characters in the figure to English.
12. Figure 10, please show the slope and intercept values of the linear curve to provide more evidence of a good fitting between the true moisture and the predicted moisture
Author Response
Dear reviewer:
Thank you for your comments and suggestions on our manuscript, and we have made changes a lot to our manuscript which you mentioned and suggested us re-writing. The following is our reply to your comments:
1. Line 94, change the “different gradients of rice grain moisture” to “rice grain of a gradient of moisture level”.
The sentence has been changed.
2. Line 97, change “divided in duplicate” to “divided into duplicate”
The sentence has been changed.
3. Line 98, change “there were 90 samples in total” to “A total of 90 samples were prepared”. This should be a separate sentence.
The sentence has been changed.
4. Line 122, remove “In this study”
It has been removed.
5. Line 131, change “then” to “and then”
The sentence has been changed.
6. Section 2.3, the author should indicate the specific method used for cross-validation and selection of samples for prediction in the Materials and Methods section.
The specific method used for cross-validation and selection of samples for prediction has been added in the Materials and Methods section.
7. Line 187-188, table 2 and figure 2. In the text, the author mentioned the model performance of the three bands, but the figure 2 and table 2 only showed the results of the chemometric models for the two bands ranging 1100-1250 nm and 1350-1550 nm. Please correct this.
The mistakes have been modified.
8. Lin 194-203, how is your results compare with available publications? Please include some discussion into this paragraph.
The comparison has been added in the discussion.
9. Figure 3 and line 206-213, could you please explain why a quadratic regression is fitted instead of a linear regression? Could this result in overfitting of the model? Also, as indicated, the PLSR model using the full spectrum provided the best quantification ability, why not use the model of the full spectra for the further application?
The fitting of 1450 nm in this paper has been modified to a linear regression equation. And now the overfitting problem has been solved. Although PLS model has good effect, the purpose of this paper is to find the characteristic band of grain moisture. Therefore, the sensor should be developed based on the relationship between grain moisture at 1450 nm and near infrared reflectance.
10. Figure 4, please add the path of the laser beam and the reflected beam in the figure.
The schematic diagram and components of sensor probe has been rewritten.
11. Figure 8, please translate the Chinese characters in the figure to English.
The Chinese characters in the figure has been translated to English.
12. Figure 10, please show the slope and intercept values of the linear curve to provide more evidence of a good fitting between the true moisture and the predicted moisture
The slope and intercept values of the linear curve has been added in Figure 10. Thank you again for the comments and suggestions. If you have any questions, we hope you can contact us.
Best wishes
Lei Lin

Reviewer 2 Report
The subject of the paper is interesting, and it is adequately motivated. However the paper itself has several flaws.
Apparently, only one type of rice was used in experiments, thus variance due to different botanical origins is not considered. If this is not the case, it should be explicitly stated in the paper.
Moreover, several prediction method are compared, but no validation statistic are given for the method of choice (1450 nm parabolic model). This is rather odd.
Beside this, explanations of methods are unclear in several points. For all these reasons I suggest a major revision of the paper.
Here follows a detailed list of points requiring revision.
Par. 2.1
Were all rice samples collected in Zuniy city of the same quality? Have you tested the sensor on different types of rice?
Par. 2.2
At lines 107-108, the authors say that the spectrometer itself carries a light source that can be placed in the photosensitive window to directly obtain the “reflection” spectrum. Do they mean “reference” spectrum, instead?
Par. 2.3.1
Explanation of PLS regression is a bit unclear. In particular, the authors say at line 134: “the number of latent variables of full spectra is 400”. However, “latent variables” is just another name of PLS factors (or “principal components” as the authors call them) and 400 is definitely a too high number for them. Maybe they refer to the original variables instead (i.e. the wavelengths).
Par. 2.3.2
Explanation of CARS is a bit unclear too. It is true that reference are given, but a more detailed explanation could be beneficial to the paper. Here are some points that should be treated the paper:
· What is the starting variable set of CARS algorithm? The full spectrum?
· The authors say that random calibration samples are sorted to test each variable set “iteratively”. Do they mean that more iterations, on different samples, are made for the same variable set? Alternatively, are both samples and variables updated at each iteration?
Par. 2.3.3
Using both RPD and R2 coefficient looks redundant, because both these parameters compare RMSE with standard deviation of target variable. Indeed R2 = 1 – (1/RPD)2.
Par. 3.1.1
Figure 1 show a sensible baseline shift in reflectance spectra. Do it depend only from moisture or is influenced by other factors? Did authors apply any pre-processing step to remove it (baseline correction, derivation, etc.)?
Rp2 coefficients reported at lines 197-199 are not coherent with those in Table2. Those relative to 1100-1250 nm band were exchanged with those of 1350-1550 nm band.
In Figure 3, moisture is predicted using “spectral absorption intensity”. How is it calculated? Values are too high to be absorbance values (A = log 1/R). Minimum reflectance in Figure 1 is about 10% and it means absorbance = 1. Even if natural logarithms were used, instead of decimal, maximum of scale should not exceed 2.3. The scale runs until 40 instead.
It should be also noted that moisture is given fraction of unit here, while is give in percent in Figures 9 and 10. The use of the same unit of measure would be advisable.
While both calibration and validation statistics were given for PLS and CARS, the same was not done for the 1450 nm parabolic fit. Only one R2 coefficient is given and it looks the calibration one. Considering that this is the method of choice, it is a serious lack. Was this method validated? How much are worth RMSEC and RMSEP?
Author Response
Dear reviewer:
Thank you for your comments and suggestions on our manuscript, and we have made changes a lot to our manuscript which you mentioned and suggested us re-writing. The following is our reply to your comments:
1. Apparently, only one type of rice was used in experiments, thus variance due to different botanical origins is not considered. If this is not the case, it should be explicitly stated in the paper.
In this paper, we carried out experiments on a kind of rice grain.
2. Moreover, several prediction method are compared, but no validation statistic are given for the method of choice (1450 nm parabolic model). This is rather odd.
The model performance of 1450 nm NIR spectra modeled by SLR have been added in Figure 2 and Table 2. And the RMSEC and RMSEP have been added.
3. Par. 2.1
Were all rice samples collected in Zuniy city of the same quality? Have you tested the sensor on different types of rice?
In this paper, we carried out experiments on a kind of rice grain.
4. Par. 2.2
At lines 107-108, the authors say that the spectrometer itself carries a light source that can be placed in the photosensitive window to directly obtain the “reflection” spectrum. Do they mean “reference” spectrum, instead?
What we want to express in this article is that the spectrum can be obtained from the light source carried inside the instrument. We have made some modifications in this paper.
5. Par. 2.3.1
Explanation of PLS regression is a bit unclear. In particular, the authors say at line 134: “the number of latent variables of full spectra is 400”. However, “latent variables” is just another name of PLS factors (or “principal components” as the authors call them) and 400 is definitely a too high number for them. Maybe they refer to the original variables instead (i.e. the wavelengths).
Thank you for your valuable comments and suggestions. The correct answer should be the original variables.
6. Par. 2.3.2
Explanation of CARS is a bit unclear too. It is true that reference are given, but a more detailed explanation could be beneficial to the paper. Here are some points that should be treated the paper:
What is the starting variable set of CARS algorithm? The full spectrum?
In this paper, the CARS models based on 950-1650 nm, 1100-1250 nm and 1350-1550 nm are established.
7. The authors say that random calibration samples are sorted to test each variable set “iteratively”. Do they mean that more iterations, on different samples, are made for the same variable set? Alternatively, are both samples and variables updated at each iteration?
It means that more iterations, on different samples, are made for the same variable set.
8. Par. 2.3.3
Using both RPD and R2 coefficient looks redundant, because both these parameters compare RMSE with standard deviation of target variable. Indeed R2 = 1 – (1/RPD)2.
The RPD has been deleted in the manuscript.
9. Par. 3.1.1
Figure 1 show a sensible baseline shift in reflectance spectra. Do it depend only from moisture or is influenced by other factors? Did authors apply any pre-processing step to remove it (baseline correction, derivation, etc.)?
In this paper, the spectra are all original spectra. Although there is baseline drift in the spectra, the experimental results show that the near infrared spectra of grain moisture with different moisture content are different, and the PLS and CARS models established in this paper have better effect.
10. Rp2 coefficients reported at lines 197-199 are not coherent with those in Table2. Those relative to 1100-1250 nm band were exchanged with those of 1350-1550 nm band.
This mistake has been modified.
11. In Figure 3, moisture is predicted using “spectral absorption intensity”. How is it calculated? Values are too high to be absorbance values (A = log 1/R). Minimum reflectance in Figure 1 is about 10% and it means absorbance = 1. Even if natural logarithms were used, instead of decimal, maximum of scale should not exceed 2.3. The scale runs until 40 instead.
The abscissa in the Figure 3 is the reflection intensity of the spectrum, and the unit of water content in the ordinate is %. The corresponding content has been revised in the text.
12. It should be also noted that moisture is given fraction of unit here, while is give in percent in Figures 9 and 10. The use of the same unit of measure would be advisable.
Now, the units of the graphs in this paper are consistent.
13. While both calibration and validation statistics were given for PLS and CARS, the same was not done for the 1450 nm parabolic fit. Only one R2 coefficient is given and it looks the calibration one. Considering that this is the method of choice, it is a serious lack. Was this method validated? How much are worth RMSEC and RMSEP?
The model performance of 1450 nm NIR spectra modeled by SLR have been added in Figure 2 and Table 2. And the RMSEC and RMSEP have been added.
Thank you again for the comments and suggestions. If you have any questions, we hope you can contact us.
Best wishes
Lei Lin

Round 2
Reviewer 2 Report
The authors answered satisfactorily to my questions but two small points. Therefore I judge the paper worth of publication, but I have to ask for a further minor revision. the two points are the following:
Par. 2.3.2, CARS, iterative modelling
The authors answered satisfactorily in their notes, but left the text unchanged in the paper. I think that the paper should clearly specify that the same variable set was repeatedly tested. This bring an additional point: if you have several models for the same set, you have several PLS regression coefficients for each variable. So, when you decide about retaining or discarding a variable, what value you consider? The mean, the maximum or the minimum?
Par. 3.1.1, exchanged Rp^2 coefficients
the authors adjusted coherently the Rp^2 coefficients, but did not change accordingly, the evaluation of models. The prediction of 1100-1250 nm is still considered the "worst" although its coefficients are higher than those of 1350-1550 nm, which is considered " better".
Although I appreciate the work of authors, I would encourage them in considering different types of rice in their future work. If they have different reflection properties it could interfere with reflectance at 1450 nm. In this case a second, reference, wavelength could be necessary for getting reliable predictions.
Author Response
Dear reviewer:
Thank you for your comments and suggestions on our manuscript, and we have made changes a lot to our manuscript which you mentioned and suggested us. The following is our reply to your comments:
The authors answered satisfactorily in their notes, but left the text unchanged in the paper. I think that the paper should clearly specify that the same variable set was repeatedly tested. This bring an additional point: if you have several models for the same set, you have several PLS regression coefficients for each variable. So, when you decide about retaining or discarding a variable, what value you consider? The mean, the maximum or the minimum?
The paper has been clearly specified that the same variable set was repeatedly tested. CARS Calculates the RMSECV value of the model and selects the minimum value corresponding to the subset of variables for the optimal subset of variables.
The authors adjusted coherently the Rp^2 coefficients, but did not change accordingly, the evaluation of models. The prediction of 1100-1250 nm is still considered the "worst" although its coefficients are higher than those of 1350-1550 nm, which is considered " better".
The errors in the article have been corrected. As a result, the 950-1650 nm containing the most comprehensive information achieved the best prediction effects ( =0.970 for PLS and =0.977 for CARS) while the prediction accuracy for 1100-1250 nm was the worst (=0.844 for PLS and =0.883 for CARS) .For 1350-1550 nm, the better prediction accuracy(=0.923 for PLS and =0.917 for CARS) was obtained compared with 1100-1250 nm, indicating that 1350-1550 nm was more sensitive to rice grain moisture in NIR region.
Thank you again for the comments and suggestions. If you have any questions, we hope you can contact us.
Best wishes
Lei Lin